# Simplifying Self-Supervised Object Detection Pretraining

## Abstract

Object detectors are often trained by first training the backbone in a self-supervised manner and then fine-tuning the whole model on annotated data. An unsupervised detector pretraining stage can also be interleaved, further improving the final performance and facilitating convergence during the supervised fine-tuning stage. However, existing unsupervised pretraining methods typically rely on low-level information to create pseudo-proposals that the model is then trained to localize, and ignore high-level class membership. The absence of class semantics from the pretraining objective causes a task gap between the pretraining and the downstream scenario, where detection is class-aware (e.g. given an image of a chair, the detector's task is to *both* localize it and assign the "chair" class to the corresponding bounding box). This gap results in suboptimal detector pretraining. We propose a framework that better aligns the pretraining and downstream stages. It consists of three simple yet key ingredients: (i) richer, semantics-based initial proposals derived from high-level feature maps, (ii) discriminative training using object pseudo-labels produced via clustering, (iii) self-training to take advantage of the improved object proposals learned by the detector. We report two main findings: (1) Our pretraining outperforms previous works on the full and low data regimes by significant margins across detector architectures. (2) We show we can pretrain detectors from scratch (including the backbone) directly on complex image datasets like COCO, paving the path for unsupervised representation learning using object detection directly as a pretext task. Code will be released.

## 1 Introduction

Object detection has been a major challenge in computer vision and the focus of extensive research efforts. Two distinct avenues of research have recently led to several breakthroughs: a) more powerful detector architectures, such as the end-to-end single stage DETR (Carion et al., 2020) family of detectors, and b) unsupervised pretraining, which leverages vast amounts of unlabeled data to improve their performance on downstream tasks where annotations are expensive, ambiguous, and/or imprecise. Notably, despite the success of unsupervised pretraining, most existing methods focus on the backbone model and neglect the detector.

Existing detector pretraining methods largely focus on DETR-based detectors due to their sample inefficiency (i.e. they require large amounts of annotated data) and slow training convergence, which means they stand to benefit the most from the unsupervised pretraining. Typically, unsupervised detector pretraining methods generate object proposals (bounding boxes or segmentation masks) randomly (e.g. Dai et al. (2021)), through heuristic-based methods such as Selective Search (e.g. Bar et al. (2022)), or with unsupervised localization techniques (e.g. Wang et al. (2023)). The pretraining task is to localize said proposals and distinguish object vs no-object regions. Thus, while the downstream task (detection) requires *both* the localization and the classification of the objects, the latter is neglected during the detector pretraining. In fact, most of the current detector pretraining methods exhibit large performance degradation when unfreezing the backbone, highlighting the task misalignment problem and also preventing the joint pretraining of the detector and backbone.

In this work, we propose SEER, a simple framework for self-supervised object detection pretraining that addresses these limitations and is, to the best of our knowledge, the first truly end-to-end detection pretraining framework, capable of effectively training the entire detector architecture (backbone

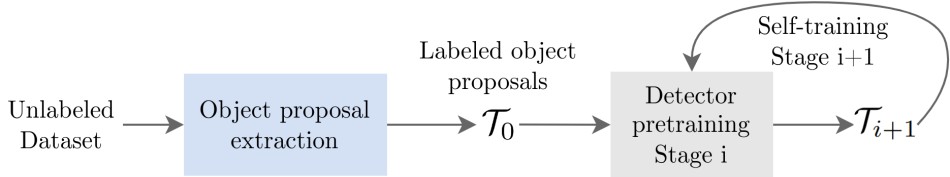

Figure 1: **SEER overview: (i)** Object proposals are extracted from images in an unsupervised manner and assigned pseudo-labels via clustering; **(ii)** The pseudo-labeled object proposals are used to train the detector, which learns to localize objects and discriminate their pseudo-class label; **(iii)** The detector then generates a new set of proposals and pseudo-labels, which are used for self-training. Best seen in color.

and detector head) jointly and even from scratch, on scene-centric images (e.g. COCO). Our method, seen in Fig. 1, has three main components, which we show *are all needed* for highly effective pre-training:

**(i) Unsupervised proposal extraction from high-level feature maps:** We obtain proposals based on high-level semantic content by clustering the feature maps produced by a self-supervised pre-trained backbone and processing the clusters to produce proposal masks. We avg-pool the resulting masks to obtain an embedding for each proposal, capturing high-level semantics.

**(ii) Detector pretraining with pseudo-labels:** The per-proposal high-level semantic embeddings are then clustered across the dataset. Cluster membership acts as pseudo-class labels. We then use the proposals of (i) and pseudo-class labels of (ii) as training data. This effectively combines localization and discrimination and achieves much better alignment with the downstream task.

**(iii) Iterative self-training:** We observe that the detector resulting from (ii) can produce better proposals than the ones it was trained on. We find that detection pretraining can be applied in an iterative fashion, where the current pretrained model produces the pseudo-labels to train itself further with improved supervision.

We conduct extensive experiments with several detector architectures and report two main findings:

**(1) Improved detection & segmentation accuracy:** We show that SEER consistently outperforms previous works by significant margins, across architectures, and in all standard benchmarks and settings for unsupervised detector pretraining.

**(2) Self-supervised representation learning from complex images:** We show that SEER can be used to train the whole network (detector head and backbone jointly) from scratch directly on complex images, demonstrating impressive performance for unsupervised representation learning.

## 2 RELATED WORKS

**Unsupervised object detector pretraining:** Object detector pretraining methods aim to pretrain the detector architecture, in addition to the backbone. Previous work in this area has mostly focused on DETR detectors, which can achieve great performance but exhibit sample inefficiency and slow convergence relative to other architectures. Thus, detector pretraining (as opposed to backbone-only pretraining) is an important task for such methods. Among these, UP-DETR (Dai et al., 2021) proposed randomly selecting areas from each image, extracting feature representations, and injecting them to the DETR detector's queries. The detector was then trained to localize the areas to which the injected representations corresponded. DETReg (Bar et al., 2022) subsequently used Selective Search (Uijlings et al., 2013) to generate object proposals as annotations for the detector. The detector was trained both to localize the proposals and represent them mimicking a pretrained backbone encoder. JoinDet (Wang et al., 2022c) improved upon DETReg by replacing Selective Search with a dynamic object proposal method that inferred the location of objects from the detector's internal activations. Siamese DETR (Huang et al., 2023) used instead a student-teacher multi-view architecture for pretraining where, in addition to class-agnostic localization, the detector is trained to learn transformation-invariant representations at the global (image) and local (object) level. Finally,

SeqCo-DETR (Jin et al., 2023) proposed sequence consistency as a pretext task, combined with a masking strategy. Notably, all of these works freeze the detector's backbone encoder during pretraining, as they suffer performance drops otherwise (Dai et al., 2021; Bar et al., 2022). This is a significant limitation, as it prevents true end-to-end self-supervised training, and makes such frameworks heavily dependent on the quality of the pretrained backbone. Beyond DETR-based detectors, CutLER (Wang et al., 2023) leveraged Wang et al. (2022b) to generate object proposals, and multiple rounds of self-training with copy-pasting (Dwibedi et al., 2017) to pretrain Mask R-CNN detectors in an unsupervised manner, demonstrating promising performance. Importantly, all of these works uniformly pretrain detectors in *a class-unaware manner*, with most relying on auxiliary objectives to improve the detectors' discriminative capacity. This creates a misalignment between the pretraining task and the downstream task of class-aware object detection, which limits the pretraining's effectiveness.

*Summary of differences with the above works:* Our method is the only one which has **all 5 following features**: (1) uses both localization and pseudo-class prediction for detector pretraining; (2) extracts a rich and varied set of object proposals from high-level semantic information to facilitate effective pretraining; (3) uses self-training to critically improve pretraining; (4) is shown capable of training both the backbone and the detector head in an end-to-end manner and, more importantly, *even from scratch*; and (5) is shown to be applicable to both two-stage (i.e. Cascade Mask R-CNN) and one-stage architectures (i.e. DETR-based).

**Unsupervised backbone pretraining for dense prediction:** Most works on unsupervised pretraining focus on pretraining the network backbone, rather than the full object detection network(Xie et al., 2021d; Hénaff et al., 2021; Wei et al., 2021; Van Gansbeke et al., 2021b; Wang et al., 2021; Huang et al., 2022; Gokul et al., 2022; Xie et al., 2021a; Wen et al., 2022; Hénaff et al., 2022; Bai et al., 2022; Karlsson et al., 2021; Islam et al., 2023; Ding et al., 2022; Li et al., 2022; Xie et al., 2021b). Specifically, works in this area do not include a localization component (i.e. they do not localize objects in images) and typically only pretrain the backbone focusing solely on representation learning. They are, therefore, distinct from unsupervised detector pretraining works, which train the detector and include a localization task, while often using pretrained backbones as initialization.

**Unsupervised object localization:** Different from object detector pretraining, this task aims to localize all objects in an image in an unsupervised manner, without considering any class information. (Van Gansbeke et al., 2022; Siméoni et al., 2021; Wang et al., 2022b; Siméoni et al., 2022; Melas-Kyriazi et al., 2022; Wang et al., 2022a). We emphasize that the main goal of these works is object localization/discovery, not the training of powerful detectors. Accordingly, the detectors trained by these works typically are not evaluated by finetuning with annotated data. Such methods also typically restrict their proposals to the most confident few (often just one) to avoid false positives, which is not well suited for detector pretraining, where training benefits from a rich set of object proposals covering as many objects (or object parts) as possible, not only the few most prominent ones. We validate this in our experiments where we outperform the state-of-the-art in unsupervised object localization Wang et al. (2023).

## 3 METHOD

Our method aims to simplify and better align the pretraining with respect to the downstream task (class-aware detection). To this end, we produce object proposals in the form of *bounding box and pseudo-class label* pairs in an unsupervised manner and then employ a self-training strategy to pretrain and iteratively refine the detector.

### 3.1 IMPROVED OBJECT PROPOSALS

Existing works either generate a very limited initial set of proposals to facilitate high precision, or use methods like Selective Search (Uijlings et al., 2013) that can generate many proposals by relying on low-level priors such as color and texture. While the former is sub-optimal due to the weaker supervisory signal, the latter is also sub-optimal for generating meaningful pseudo-class labels since it does not capture high-level semantics. Our aim is to address this gap by utilizing semantic information from self-supervised image encoders to produce rich object proposals and coherent pseudo-class labels.

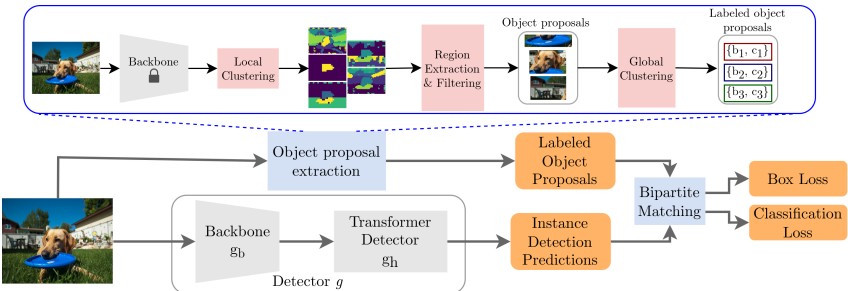

Figure 2: Overview of SEER's pretraining Stage 1. Pseudo-labeled region proposals are extracted at the start of training leveraging a self-supervised pretrained backbone. Those proposals are then used to train the detector to both localize objects within the image, and to discriminate their pseudo-labels. Best seen in color.

Specifically, we extract feature maps using a pretrained self-supervised encoder and leverage a bi-level clustering strategy. The first level (termed local clustering) results in bounding box proposals and associated feature representations. The second level, termed global clustering, uses cluster membership to assign a pseudo-class label to each proposal. Our method leads to rich and diverse region proposals and is essential for the state-of-the-art results of SEER, which we discuss in detail in Appendix D.

**Unsupervised proposal extraction:** Given an input image $X \in \mathbb{R}^{3 \times H \times W}$, we use a self-supervised pretrained encoder to extract feature maps $\mathbf{F}_l \in \mathbb{R}^{d_l \times H_l \times W_l}$ from each of the encoder's levels $l$. Given a feature map $\mathbf{F}$, we employ pixel-wise clustering to group semantically similar features (local clustering). This results in a set of masks $\mathbf{M} = \{\mathbf{m}_k\}_{k=1:K}$, where $K$ represents the number of clusters, which is a user-defined parameter. In order to provide good coverage for all objects in the image, we apply clustering with different values $K \in \mathcal{K}$ and use feature maps from different layers $l \in \mathcal{L}$, leading to a set of masks $\mathbb{M} = \bigcup \{\mathbf{M}^{l,K}\}_{K \in \mathcal{K}, l \in \mathcal{L}}$.

Next, the different connected components of each mask are computed, leading to a set of regions $\mathbb{R}$. Each region $\mathbf{r} \in \mathbb{R}$ is then used to extract a bounding box (proposal) $b$ and a corresponding feature vector $f$, where $f$ is computed by average-pooling the last layer feature map $\mathbf{F}_L$ over $\mathbf{r}$.

**Proposal filtering:** Due to the clustering at multiple levels of the encoder, the process leads to noisy and overlapping proposals. We employ a number of filters to refine them, such as merging proposals that have a high IoU and proposals with highly related semantic content. This results in a set of $N(i)$ bounding box-feature vector pairs for image $i$, $\{b_n, f_n\}_{n=1}^{N(i)}$.

**Pseudo-class label generation:** We then cluster proposals across the whole dataset (global clustering) based on the feature vectors, i.e. we perform a single clustering round on $\{f_n^i\}_{n=1:N(i)}^{i=1:I}$, obtaining clusters $S_c$ for $c \in \{1, .., \mathcal{C}\}$. This results in a training set $\mathcal{T}_0 = \{X_i, \{(b_n^i, c_n^i)\}\}$, where $c_n^i$ is defined by cluster membership, i.e. $f_n^i \in S_{c_n^i}$.

We used Spectral Clustering (Ng et al., 2001) for the local clustering and K-Means for global clustering. While Spectral Clustering usually performs better, it cannot handle the millions of data points involved in the global clustering step, since the memory requirements are quadratic with respect to the number of data points. However, any clustering algorithm may be used in either case.

## 3.2 PRETRAINING AND SELF-TRAINING

We can now use the training set $\mathcal{T}_0$ to train an object detector. In particular, given an input image and its corresponding extracted object proposals $y$, the network predicts a set $\hat{\mathbf{y}} = \{\hat{y}_q\}_{q=1}^{Q}$, where $\hat{y}_q = (\hat{b}_q, \hat{c}_q)$ comprises the predicted bounding box and predicted category. We note that the extracted proposals $y$ are padded to size $Q$ with $\varnothing$ (no object). We emphasize that SEER is compatible with *any detector architecture*, as we train the detector on simple class-aware detection. Here, for ease of notation and without loss of generality, we assume a DETR-based detector. The ground truth and

the predictions are put in correspondence via bipartite matching, formally defined in Eq. (1), where $\mathfrak{S}_Q$ is the space of permutations of $Q$ elements. The loss between $\mathbf{y}$ and $\hat{\mathbf{y}}$ is computed in Eq. (2), as a combination of a bounding box matching loss and a class matching loss:

$$\hat{\sigma} = \underset{\sigma \in \mathfrak{S}_Q}{\arg\min} \sum_q^Q \mathcal{L}(y_q, \hat{y}_{\sigma(q)}) \tag{1}$$

$$\sum_{q=1}^Q \left( -log\hat{p}_{\hat{\sigma}(q)}(c_q) + \mathbf{1}_{\{c_q \neq \varnothing\}} \mathcal{L}_{box}(b_q, \hat{b}_{\hat{\sigma}(q)}) \right), \tag{2}$$

where $\hat{p}$ indicates the predicted per-class probabilities. The indicator function $\mathbf{1}_{c_i \neq \varnothing}$ represents that the box loss only applies to predictions that have been matched to object proposals $y$. Minimizing this loss results in weights $\Theta_0$.

Upon training the detector in this way, we observe that it can identify more objects than those in our original proposals. Critically, this includes smaller and more challenging objects, which can contribute to a stronger supervisory signal. We thus generate a new set of pseudo-labels for image $i$ as $\{g(X_i; \Theta_0)\}$, where $g = (g_b, g_h)$ are the detection network, backbone and head respectively. It is typical during self-training that object proposals are filtered through a confidence threshold (Wang et al., 2022a; 2023). We find, however, that in our case this leads to the removal of small or challenging instances such as partially occluded or uncommon objects. Thus, we instead filter the top-100 proposals of the detector for overlap only, so that any two boxes have an IOU lower than 0.55 (following (Solovyev et al., 2021)), with only the most confident box being kept when such conflicts exist. This leads to a training set $\mathcal{T}_1$.

A new set of weights $\Theta_i$ can be obtained by using training set $\mathcal{T}_i$ and using $\Theta_{i-1}$ to initialize the weights. Simultaneously, $\Theta_i$ can be used to generate a new training set $\mathcal{T}_{i+1}$. While this process can be iterated indefinitely, we notice optimal performance involves just two rounds of training, which we refer to as Stages 1 & 2. Stage 1 training, including the proposal extraction process for $\mathcal{T}_0$ is shown in Fig. 2.

We highlight that, importantly, the proposed pretraining is very well-aligned with the downstream task, i.e. supervised class-aware object detection, and it allows the pretraining of *both* the backbone and the detection head simultaneously. This is unlike other detector pretraining methods (Dai et al., 2021; Bar et al., 2022; Wang et al., 2022c) that require freezing the backbone to avoid performance degradation.

The whole method is summarized in Algorithm 1 of Appendix E.

## 4 EXPERIMENTAL SETTING

We apply SEER to two DETR-based architectures (Deformable DETR (Zhu et al., 2021) and ViDT+ (Song et al., 2022)) and an R-CNN architecture (Cascade Mask R-CNN (Cai & Vasconcelos, 2018)), focusing on the former, as DETR's end-to-end single-stage architecture typically performs better and is better suited for representation learning. In order to compare with prior work on object detection pretraining, we follow Bar et al. (2022) for Def. DETR and ViDT+, and Wang et al. (2023) for Cascade Mask R-CNN in terms of datasets, hyperparameters and experiments. For unsupervised representation learning, in the absence of a predefined protocol, we use the ViDT+ detector and experiment with the most well-established datasets in object detection. The hyperparameters for each experiment are provided in detail in Appendix A. Unless stated otherwise, for methods other than SEER we report results from the respective papers, except where ViDT+ is used.

**Datasets:** We use the training sets of ImageNet (Russakovsky et al., 2015), Open Images (Krasin et al., 2017) and MS COCO (Lin et al., 2014) for unsupervised pretraining. For finetuning (with annotations) we use the training sets of MS COCO and PASCAL VOC (Everingham et al., 2010). Results are reported for the corresponding validation sets, using the Average Precision (AP) and Average Recall (AR). Details on the datasets are provided in Appendix B.

Table 1: **Object detection results on COCO.** Methods are pretrained on ImageNet, finetuned on MS COCO `train2017` and evaluated on `val2017`. 1: Backbone initialized with MoBY and pretrained with SEER (pretrained detection head was discarded).

| Detector | Backbone Pretraining | Detector Pretraining | Frozen Backbone | AP |
|---|---|---|---|---|
| Cascade Mask R-CNN (Cai & Vasconcelos, 2018) | DINO | - | ✗ | 44.4 |
| | | CutLER Wang et al. (2023) | ✗ | 44.7 |
| | | **SEER** | ✗ | **45.0** |
| Def. DETR (Zhu et al., 2021) | SwAV | - | - | 45.2 |
| | | UP-DETR (Dai et al., 2021) | ✓ | 44.7 |
| | | DETReg (Bar et al., 2022) | ✓ | 45.5 |
| | | JoinDet (Wang et al., 2022c) | ✓ | 45.6 |
| | | SeqCo-DETR (Jin et al., 2023) | ✓ | 45.8 |
| | | Siamese DETR (Huang et al., 2023) | ✓ | 46.3 |
| | | **SEER** | ✗ | **46.7** |
| ViDT+ (Song et al., 2022) | MoBY | - | - | 48.3 |
| | **SEER**[1] | - | - | **48.8** |
| | MoBY | DETReg | ✓ | 49.1 |
| | MoBY | DETReg | ✗ | 47.8 |
| | MoBY | **SEER** | ✗ | **49.6** |

**Architectures:** For our experiments we use Def. DETR, ViDT+ and Cascade Mask R-CNN. Def. DETR and Cascade Mask R-CNN are primarily used to compare with prior work for detector pretraining. ViDT+ is currently one of the best-performing DETR-based object detection methods, with a highly efficient architecture that unifies the backbone and encoder components of the DETR framework. It is, therefore, further used to compare against unsupervised representation learning methods. Following Bar et al. (2022); Wang et al. (2023), Def. DETR Cascade and Mask R-CNN detectors use ResNet-50 (He et al., 2016) backbones initialized with SwAV (Caron et al., 2020) and DINO (Caron et al., 2021) respectively. ViDT+ uses a Swin-T (Liu et al., 2021) backbone initialized with MoBY (Xie et al., 2021c), unless stated otherwise. In all cases, the backbones were trained in a fully unsupervised manner on ImageNet.

## 5 EXPERIMENTS

We highlight two main results, namely state-of-the-art results for detection pretraining and competitive results for self-supervised representation learning for detection, including pretraining on scene-centric data such as COCO and OpenImages **from scratch**. We complement these results with a comprehensive set of ablation studies.

### 5.1 OBJECT DETECTION PRETRAINING

We evaluate SEER following the standard protocol for object detection pretraining, as defined by Bar et al. (2022) for DETR-based architectures and Wang et al. (2023) for Cascade Mask R-CNN, which include experiments in the full-data, semi-supervised and few-shot settings.

**Full data setting:** We provide a comprehensive set of comparisons with competing detector pretraining methods in Tab. 1, where we pretrain 3 detector architectures on ImageNet, finetune on COCO `train2017` and evaluate on `val2017`. We also report results for ImageNet pretraining and PASCAL VOC finetuning with Def. DETR in Tab. 2. As Tables 1 and 2 show, our method significantly outperforms competing detector pretraining methods across datasets and with all 3 detector architectures. Interestingly, all prior work on DETR pretraining requires freezing the backbone. We quantitatively assess the impact of this requirement by making the DETReg backbone trainable, and observe steep performance degradation. Contrary to all these works, SEER supports a trainable backbone due to its better alignment of the pretraining and downstream tasks.

Table 2: **Object detection results on PASCAL VOC**. Methods were pretrained on ImageNet, fine-tuned on PASCAL VOC `trainval07+2012` and evaluated on `test07`.

| Method | $AP$ | $AP_{50}$ | $AP_{75}$ |
|---|---|---|---|
| SwAV | 61.0 | 83.0 | 68.1 |
| DETReg | 63.5 | 83.3 | 70.3 |
| JoinDet | 63.7 | 83.8 | 70.7 |
| SeqCo-DETR | 64.1 | 83.3 | 70.3 |
| **SEER** | **64.8** | **84.6** | **72.7** |

Table 3: **Semi-supervised results against detector pretraining methods.** Following Bar et al. (2022), Def. DETR detectors were pretrained on MS COCO `train2017`, finetuned on k% labeled samples, and evaluated on `val2017`.

| Method | AP | | | |
|---|---|---|---|---|
| | 1% | 2% | 5% | 10% |
| SwAV | 11.79±0.3 | 16.02±0.4 | 22.81±0.3 | 27.79±0.2 |
| DETReg | 14.58±0.3 | 18.69±0.2 | 24.80±0.2 | 29.12±0.2 |
| JoinDet | 15.89±0.2 | - | - | 30.87±0.1 |
| **SEER** | **18.19±0.1** | **21.80±0.2** | **26.90±0.2** | **30.97±0.2** |

**Semi-supervised setting:** We present results in Tab. 3 for Def. DETR, pretrained on COCO `train2017` and fine-tuned on k% labeled samples, following Bar et al. (2022). In Tab. 4 we compare with works focusing on unsupervised localization following Wang et al. (2023), where we pretrained a Cascade Mask R-CNN on ImageNet and fine-tuned on COCO `train2017` with k% samples, including instance segmentation results. In both cases, SEER outperforms previous works by large margins, particularly in the more challenging settings with fewer labeled samples. Notably, despite our pretraining being focused on detection, our method outperforms FreeSOLO and CutLER in segmentation performance as well, which highlights its effectiveness.

**Few-shot setting:** We follow protocol defined in Bar et al. (2022) and pretrain Def. DETR on ImageNet and report results for two settings: a) further pre-training on COCO `train2014` with 60 base classes, and then fine-tuning in a few-shot setting with $k \in \{10, 30\}$ instances from all classes, b) we skip further training on the base classes. Results are reported in Tab. 5 on the novel classes of `val2014`, and demonstrate that SEER not only outperforms DETReg by significant margins. Furthermore, SEER's performance without base class finetuning is very close to its performance with it. These results support that a) our method drastically reduces detector architectures' dependency on annotated data, and b) SEER's learned representations are already class-aware, and the pseudo-labels produced by our method are good enough that SEER can align with COCO's classes with minimal (10-shot) supervision. We conduct a more in-depth analysis of the few-shot setting outcomes in Appendix C.

## 5.2 SELF-SUPERVISED REPRESENTATION LEARNING ON SCENE-CENTRIC IMAGES

In this section we examine SEER's performance on scene-centric data, and its ability to learn self-supervised representations (i.e., train a backbone) suitable for detection. We begin by validating that SEER, when trained on scene-centric data (e.g. COCO), can perform competitively compared to ImageNet pretraining. Then we use SEER directly for self-supervised representation learning on scene-centric data (i.e., training from scratch on COCO/Open Images), showing promising results. Finally, we show that pretraining on COCO leads to representations that transfer to ImageNet under the linear-probe setting.

**Object vs Scene-centric pretraining:** In the full data experiments in Sec. 5.1 we followed prior literature: we initialized our method with a backbone pretrained on object-centric data (ImageNet) and pretrained the detector again on object-centric data. In Tab. 6, we present results for SEER when the detector is pretrained on scene-centric data instead. Specifically, we now pretrain ViDT+ on

Table 4: **Semi-supervised results against unsupervised localization methods**. FreeSOLO uses SOLOv2 (Wang et al., 2020b) and is pretrained on MS COCO `train2017+unlabeled2017`. CutLER and SEER use Cascade Mask R-CNN and are pretrained on ImageNet. All methods are finetuned on MS COCO `train2017` and evaluated on `val2017`.

| Method | AP (Box / Mask) | | | | |
| | 1% | 2% | 5% | 10% | 100% |
| --- | --- | --- | --- | --- | --- |
| FreeSOLO | - / - | - / - | - / 22.0 | - / 25.6 | - / - |
| CutLER | 16.8 / 14.6 | 21.6 / 18.9 | 27.8 / 24.3 | 32.2 / 28.1 | 44.7 / 38.5 |
| **SEER** | **20.8 / 17.5** | **25.2 / 21.2** | **30.0 / 25.5** | **33.8 / 29.0** | **45.0 / 38.8** |

Table 5: **Few-shot results**. Def. DETR detectors were pretrained on ImageNet and finetuned on $k \in \{10, 30\}$ instances from each class of MS COCO `train2014`. Results reported on the novel classes of `val2014`. DETReg results reproduced in our codebase using the official checkpoint.

| Method | Base Class Finetuning | Novel Class AP | | Novel Class AP$_{75}$ | |
| | | 10 | 30 | 10 | 30 |
| --- | --- | --- | --- | --- | --- |
| DETReg | ✗ | 5.6 | 10.3 | 6.0 | 10.9 |
| **SEER** | | **10.3** | **14.5** | **10.9** | **15.1** |
| DETReg | ✓ | 9.9 | 15.3 | 10.9 | 16.4 |
| **SEER** | | **12.4** | **18.9** | **13.1** | **20.4** |

COCO and Open Images (keeping the initialization settings described in Sec. 4), finetune on COCO `train2017` and present results on its validation set. We further report the class-unaware object detection performance in terms of average recall (AR) as it hints at different behaviors between the two settings in this regard.

We observe that, in all cases, our method improves over the baseline, including when we pretrain and finetune on the same set of data (COCO). Furthermore, we find that SEER performs similarly when trained on MS COCO relative to Open Images, despite the latter being a much larger dataset. Finally, while ImageNet leads to the best outcomes, Open Images pretraining is very close. Combined, these findings show that SEER is: a) sample efficient, achieving similar performance pretraining on COCO and on the larger ImageNet and Open Images datasets, and b) flexible, being able to handle both object-centric and scene-centric data. Tab. 6 also provides an insight as to why ImageNet pretraining performs best. As seen by contrasting AR scores, ImageNet's detector localizes more objects correctly. This indicates that the proposals generated for ImageNet are relatively better, which likely leads to better supervision, especially when self-training. Overall, these results indicate that SEER does not require carefully curated object-centric data to achieve competitive results.

**Self-supervised representation learning on scene-centric data:** Experiments conducted in previous sections initialize the backbone with weights obtained by self-supervised training on ImageNet. In this section, we evaluate the representation learning capacity of SEER by pretraining a ViDT+ detector from an *untrained* backbone (from scratch) to examine whether independent backbone pre-

Table 6: **Object-centric vs Scene-centric pretraining.** SEER is pretrained on MS COCO `train2017`, ImageNet and Open Images, finetuned on `train2017` and evaluated on `val2017`. 1: We finetune an untrained detector, initialized with a MoBY backbone.

| Detector Pretraining | AP | AP$_{50}$ | AP$_{75}$ | AR$^{100}$ |
| --- | --- | --- | --- | --- |
| -[1] | 48.3 | 66.9 | 52.4 | - |
| COCO | 49.1 | 67.8 | 53.1 | 25.1 |
| ImageNet | **49.6** | **68.2** | 53.8 | 27.1 |
| Open Images | 49.4 | 67.9 | **53.9** | 25.5 |

Table 7: **Pretraining from scratch**. We pretrain SEER *without backbone initialization*, finetune on MS COCO `train2017` and evaluate on `val2017`. For comparison, we finetune ViDT+ without any pretraining and with a MoBY-pretrained backbone.

| Backbone Pretraining | Detector Pretraining | Detector | Pretraining Dataset | AP |
|---|---|---|---|---|
| MoBY | - | | ImageNet | 47.6 |
| DetCon  (Hénaff et al., 2021) | - | FCOS* | ImageNet | 48.4 |
| Odin  (Hénaff et al., 2022) | - | | ImageNet | 48.5 |
| - | - | | ImageNet | 38.5 |
| MoBY | - | | ImageNet | 48.3 |
| - | **SEER** | ViDT+ | COCO | 48.3 |
| - | **SEER** | | Open Images | 48.8 |
| - | **SEER** | | ImageNet | **49.2** |

Table 8: **Linear probing**. We pretrain SEER with ViDT+ on MS COCO `train2017`, and apply the backbone to linear evaluation on ImageNet. Results reported on ImageNet's validation set. Results for other methods are taken from Van Gansbeke et al. (2021a).

| Backbone Pretraining | Acc |
|---|---|
| DenseCL (Wang et al., 2021) | 49.9 |
| VirTex (Desai & Johnson, 2021) | 53.8 |
| MoCo (He et al., 2020) | 49.8 |
| Van Gansbeke et al. (Van Gansbeke et al., 2021a) | 56.1 |
| **SEER** | **56.4** |

training is indeed necessary. We pretrain on object-centric (ImageNet) and scene-centric (COCO & Open Images) datasets and present results in Tab. 7. For completeness, we also provide results for other methods that focus on self-supervised backbone-only pretraining, noting that they use a different detector architecture during finetuning.

Results again show that SEER performs best with a well-curated, object-centric pretraining dataset, but is competitive even when trained on complex, scene-centric images. Specifically, SEER performs on par with backbone-only ImageNet pretraining (MoBY) when pretrained on COCO, and outperforms it when pretrained on Open Images. This outcome supports our thesis that unsupervised pretraining directly on scene-centric data with an object detection task is feasible and effective.

We further evaluate the quality of the COCO-pretrained backbone by performing a linear probe experiment on ImageNet. Tab. 8 shows SEER's performance as well as that of prior work. We note that prior work use a ResNet50 encoder, and thus a direct comparison is hard. It is however clear that our method is competitive, despite being pretrained for object detection, highlighting the natural fit of SEER for general-purpose representation learning from scene-centric images.

## 6 CONCLUSION

We have proposed SEER, a novel method for self-supervised end-to-end object detector pretraining. Compared to prior work, our method aligns pretraining and downstream tasks through the careful construction of object proposals and pseudo-labels and the use of self-training. We extensively evaluate SEER in typical object detector pretraining benchmarks and demonstrate that it consistently outperforms previous methods across detector architectures. However, unlike prior work, we show that SEER is also capable of effectively pretraining the backbone. This brings our method in line with the wider literature on self-supervised representation learning for detection. We again show competitive performance in this area and explore novel settings, specifically pretraining with scene-centric datasets and even pretraining from scratch. Overall, we believe our framework not only outperforms existing detector pretraining methods but also represents a promising step toward self-supervised, fully end-to-end object detection pretraining on uncurated images.

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

# Appendix

## A   TRAINING HYPERPARAMETERS

In this section we provide detailed hyperparameters for each training setting included in the main paper. We use three detectors, Def. DETR Zhu et al. (2021), ViDT+ Song et al. (2022) and Cascade Mask R-CNN (Cai & Vasconcelos, 2018). For Def. DETR and ViDT+ we typically follow the training settings proposed in their respective papers for finetuning and DETReg Bar et al. (2022) for pretraining. For Cascade Mask R-CNN we follow Wang et al. (2023). More specifically, unless stated otherwise, the following hyperparameters apply:

**For Def. DETR**, we train SEER following Bar et al. (2022). Specifically, we pretrain for 5 epochs per stage on ImageNet with a batch size of 192 and a fixed learning rate of 0.0002. For finetuning, we train on COCO for 50 epochs and PASCAL VOC for 100 epochs, with a batch size of 32. The learning rate is set to 0.0002, and is decreased by a factor of 10 at epoch 40 and 100 for COCO and PASCAL VOC respectively.

**For ViDT+**, we use the training hyperparameters proposed in Song et al. (2022). Specifically, unless stated otherwise, ViDT+ is pretrained for 10 epochs per stage on ImageNet and Open Images, and for 50 epochs per stage on COCO, with batch size 128. In all cases, the learning rate is set to 0.0001 and follows a cosine decay schedule.

**Cascade Mask R-CNN**, we use the pretraining and fine-tuning hyperparameters proposed in Wang et al. (2023). Specifically, unless stated otherwise, Cascade Mask R-CNN is pretrained for 160,000 steps per stage on ImageNet with batch size 16. The learning rate is set to 0.01 and decreased by a factor of 10 at after 80,000 training steps.

Unless stated otherwise, we pretrain with 2048 pseudo-classes (i.e. we set the number of clusters for the global clustering step to 2048), and apply one round of self-training, following our findings in Tab. 13. Finally, during pretraining, we use the mosaic augmentation Bochkovskiy et al. (2020).

For specific experiments conducted in the paper, we note changes relative to the settings described above:

**Full data regime:** Same as above.

**Semi-supervised:** For Def. DETR we follow DETReg Bar et al. (2022), we finetune on COCO for 2,000 epochs for 1% of samples annotated, 1,000 epochs for 2% of samples, 500 epochs for 5% of samples, and 400 epochs for 10% of samples. The learning rate is kept fixed at 0.0002. Results in Table 3 are measured over 5 runs, with different, randomly sampled annotated samples. For Cascade Mask R-CNN, we closely follow the training setting and evaluation protocol used in Wang et al. (2023).

**Few-shot:** We finetune on COCO's base classes, using the splits proposed in Wang et al. (2020a). For the standard few-shot setting we a) finetune on the base classes following the COCO finetuning settings outlined above, and b) finetune on the 10- and 30-shot sets for 30 and 50 epochs respectively, with a fixed learning rate of 0.0002 and 0.00004. For the extreme setting, we directly finetune on the 10- and 30-shot sets for 400 epochs with a learning rate of 0.0002 that is decreased by a factor of 10 after 320 epochs. Results in Table 4 correspond to the best validation score of each run during training, averaged over 5 runs, with k-shot samples corresponding to seeds 1-5 of Wang et al. (2020a). When finetuning on the k-shot instances, the backbone is kept frozen in both settings.

**Object vs Scene-centric pretraining:** Same as above.

**Self-supervised representation learning on scene-centric data:** For these experiments, where the entire architecture is initialized from scratch (backbone & detector), we train for 1,000 epochs on COCO, 100 epochs on ImageNet, and 70 epochs on Open Images. This allows for a fair comparison, with approximately the same number of training steps across datasets.

## B    DATASETS

In our paper, we use the training sets of ImageNet Russakovsky et al. (2015), Open Images Krasin et al. (2017) and MS COCO (COCO) Lin et al. (2014) for unsupervised pretraining. We use the training sets of MS COCO and PASCAL VOC Everingham et al. (2010) for supervised finetuning and their validation sets for evaluation. ImageNet includes 1.2M object-centric images, classified with 1,000 labels and without object-level annotations. Open Images includes 1.7M scene-centric images, and a total of 14.6M bounding boxes with 600 object classes. COCO is a scene-centric dataset with 120K training images and 5K validation images containing 80 classes. PASCAL VOC is scene-centric and contains 20K images with object annotations covering 21 classes.

## C    CONVERGENCE & ALIGNMENT ANALYSIS

In this section we discuss the convergence and alignment properties of SEER by analyzing the results of the "extreme" few-shot experiments. As discussed in Sec. 5, in this setting we pretrain Def. DETR on ImageNet, and then finetune directly on COCO `train2014`, using $k \in \{10, 30\}$ instances from all classes.

Table 9: Results of "extreme" few-shot training for 50 epochs and 400 epochs.

| Method | Epochs | Novel Class AP | | Novel Class AP$_{75}$ | |
| --- | --- | --- | --- | --- | --- |
| | | 10 | 30 | 10 | 30 |
| DETReg | 50 | 1.9 | 3.4 | 1.8 | 3.52 |
| **SEER** | | **8.32** | **13.9** | **8.06** | **14.4** |
| DETReg | 400 | 5.6 | 10.3 | 6.0 | 10.9 |
| **SEER** | | **10.3** | **14.5** | **10.9** | **15.1** |

In Figures 3 and 4 we present the AP scores for SEER and DETReg during training, averaged over 5 runs and measured over the validation set's novel classes. As was noted in Sec. 5, SEER outperforms DETReg by large margins. Notably, however, it is also shown to converge much faster. More specifically, in Tab. 9 we present results for 50 epochs of k-shot finetuning against the performance reached after 400 epochs. In both cases, we average the best validation score across 5 runs. We see that, at 50 epochs, SEER has already reached near-peak performance, while DETReg converges at a much slower rate.

This means SEER effectively alleviates the sample inefficiency and slow convergence of DETR architectures, and makes our method particularly useful when annotations and/or computational resources are extremely scarce. These results provide further support for our conclusions in Sec. 5,

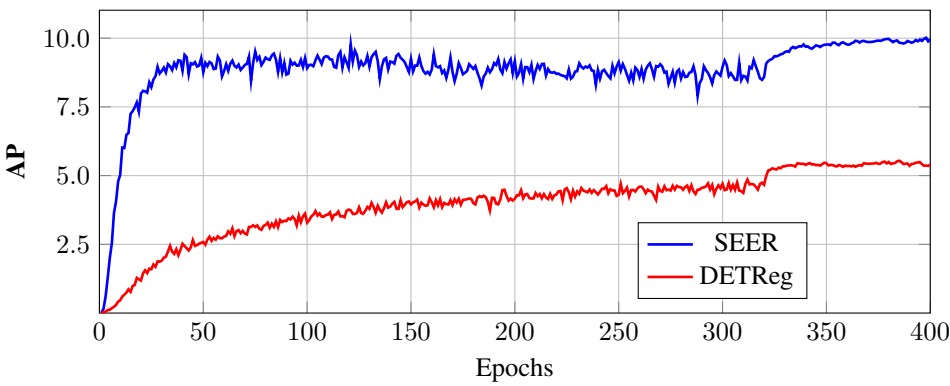

Figure 3: AP scores on COCO's `val2014` novel classes during finetuning with k=10 instances per class. Results averaged over 5 runs.

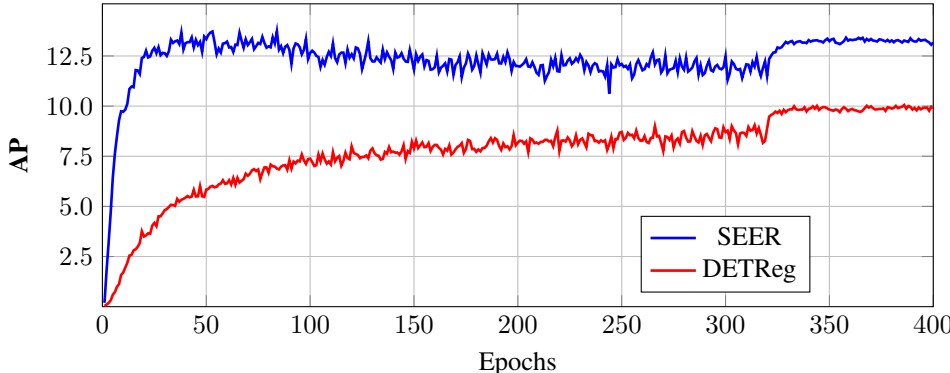

Figure 4: AP scores on COCO's `val2014` novel classes during finetuning with k=30 instances per class. Results averaged over 5 runs.

namely that SEER is much better aligned with the downstream task, with learned object representations that are well suited for class-aware object detection, so that minimal training and supervision can lead to strong performance.

# D  ANALYSIS AND ABLATIONS

Throughout this section we use ViDT+ and, unless stated otherwise, pretrain on ImageNet for 10 epochs per stage.

**Impact of object proposals:** We evaluate our object proposal method in two ways: a) we examine how well it localizes objects by computing the Average Recall (AR) score on COCO `val2017` (see Tab. 10), and b) we investigate its impact on SEER by replacing it with Selective Search, and present the outcomes (see Tab. 11).

Table 10: **Quality of proposals:** AR results on COCO `val2017`. The first section presents results for the initial extraction of object proposals, while the lower two sections present results for proposals generated by detection/segmentation architectures trained on the initial proposals.

| Object proposals | Detection Architecture | $AR^{100}$ |
|---|---|---|
| Sel. Search | - | 10.9 |
| **SEER**-St. 0 | - | **13.4** |
| DETReg | | 21.5 |
| **SEER**-St. 1 | ViDT+ | 25.9 |
| **SEER**-St. 2 | | **27.1** |
| CutLER | | **32.7** |
| **SEER**-St. 1 | Cascade Mask R-CNN | 24.5 |
| **SEER**-St. 2 | | 24.6 |

Tab. 10 includes results both for our initial proposals (noted as SEER-St. 0), and the proposals generated by pretrained detectors. Results show that our approach is superior to Selective Search and that detector pretraining significantly improves over our initial proposals, supporting our decision to self-train. We observe also that our framework leads to better localization results than DETReg. Most interestingly, we observe that CutLER performs better in terms of localization than SEER, even though SEER consistently outperforms CutLER in terms of object detection pretraining. This reinforces our claim in Sec. 2, that unsupervised localization methods generate annotations and follow training processes that are not necessarily good for detector pretraining.

In Tab. 11 we find that, using Selective Search proposals, SEER still outperforms the MoBY baseline, but we observe a performance drop relative to our object proposal method. We attribute this

Table 11: **Impact of initial proposals:** AP results on COCO `val2017`, using different initial object proposal methods.

| Method | Proposals | AP | $AP_{50}$ | $AP_{75}$ |
|---|---|---|---|---|
| MoBY | - | 48.3 | 66.9 | 52.4 |
| SEER-St. 1 | Sel. Search | 48.7 | 67.3 | 52.7 |
| SEER-St. 2 | | 48.6 | 67.1 | 52.2 |
| SEER-St. 1 | Our Anns. | 48.9 | 67.4 | 52.9 |
| SEER-St. 2 | | **49.6** | **68.2** | **53.8** |

Table 12: **Number of classes**. Pretraining and finetuning on COCO, evaluation in terms of training accuracy, AR of the pretrained detector, and AP of the finetuned model. 1 class implies class-unaware pretraining.

| Classes | ACC | AR | AP |
|---|---|---|---|
| 1 | - | **25.2** | 41.2 |
| 256 | **80.01** | 23.9 | 43.8 |
| 512 | 75.13 | 24.0 | 43.9 |
| 2048 | 53.75 | 23.9 | **44.1** |

to two reasons: a) out method likely produces more discriminative descriptors $f$ by aggregating representations over a mask of semantically related pixels, rather than over a box, which is the case for Selective Search. This, in turn, leads to better pseudo-labels. b) Our proposals are more robust (see Tab. 10), and therefore provide better supervision. In summary, we conclude that SEER is robust to different object proposal methods, but greatly benefits from an appropriate method choice.

**Number of classes:** We ablate the number of pseudo-classes produced by the global clustering of object proposals. For this set of experiments, we pretrain and finetune on COCO `train2017` for 25 epochs each. Note this is a simplified (and cheaper) setting for the purpose of ablating. We find that, during pretraining, increasing the number of classes leads to decreased training accuracy (ACC) and class-unaware AR (measured on the validation set), which is expected, since increasing the number of classes makes the task harder. However, the AP score after finetuning increases, indicating that the pretrained detector is more powerful. Overall, results indicate that our method is fairly robust to the number of clusters chosen.

**Self-training stages:** We examine the impact of self-training in Tab. 13, and find that it produces meaningful gains. We explore additional self-training with ViDT+ (Song et al., 2022), but observe no benefits, and therefore limit self-training to one round throughout the paper.

**Schedule length:** In Tab. 14 we examine the impact of a longer training schedule on our method for both training stages by extending training from 10 to 25 epochs per stage. The results show that a longer training schedule can have some beneficial, yet marginal, effect. Interestingly, Tab. 14 highlights the importance of self-training, as two training stages totaling a combined 20 epochs (10 per stage) clearly outperform a single training round of 25 epochs.

Table 13: **Self-training rounds.** AP results for ViDT+ pretrained with SEER on ImageNet and finetuned on COCO. Avg. proposals per image are measured during training.

| Detector | Stage | AP | $AP_{50}$ | $AP_{75}$ |
|---|---|---|---|---|
| ViDT+ (Song et al., 2022) | 1 | 48.9 | 67.4 | 52.9 |
| | **2** | **49.6** | **68.2** | 53.8 |
| | 3 | 49.6 | 68.0 | **53.9** |
| Def. DETR (Zhu et al., 2021) | 1 | 46.1 | 64.6 | 50.3 |
| | **2** | **46.7** | **65.4** | **50.9** |

Table 14: **Scheduler length.** AP results for varying training epochs. 10 and 25 epoch Stage 2 models are initialized from 10 and 25 epoch Stage 1 models respectively.

| Stage | Epochs | AP | $AP_{50}$ | $AP_{75}$ |
|-------|--------|------|------|------|
| 1 | 10 | 48.9 | 67.4 | 52.9 |
| 1 | 25 | 49.2 | 67.7 | 53.6 |
| 2 | 10 | 49.6 | 68.2 | 53.8 |
| 2 | 25 | 49.7 | 68.1 | 54.2 |

# E ALGORITHM

We present SEER as an algorithm in Algorithm 1.

---

**Algorithm 1** Pretraining

---

**Require:** $\{X_i\}_{i=1}^I$, Net $g = (g_b, g_h)$, initial params. $\Theta_0$

1:                                                                       ▷ Unsup. train set gen., Sec. 3.1
2: **for** $i = 1 : N$ **do**
3:      $\mathbf{F}_l \leftarrow g_b(X_i)$
4:      $\mathbb{M}_i \leftarrow \bigcup \mathrm{Cluster}(F_l, K)$                                         ▷ $K \in \mathcal{K}, l \in \mathcal{L}$
5:      $\mathbb{R}_i \leftarrow \mathrm{Connected\ Components}(\mathbb{M}_i)$
6:      $\{b_n^i, f_n^i\}_{N(i)} \leftarrow \mathrm{Filter}(\mathbb{R}_i)$
7: **end for**
8: $\{c_n^i\} \leftarrow \mathrm{K\text{-}Means}(\{f_n^i\}, K = C)$                          ▷ Pseudo-classes
9: $\mathcal{T}_0 \leftarrow \left\{ X_i, \{(b_n, c_n)\}_{n=1}^{N(i)} \right\}_{i=1}^I$
10:                                                            ▷ Self-training (Sec. 3.2)
11: **for** $j$ stages **do**
12:      $g(-; \Theta_{j+1}) \leftarrow \mathrm{Train}\,(\mathcal{T}_j, g)$                    ▷ Using Eq. (2)
13:      $\mathcal{T}_{j+1} \leftarrow \mathrm{Filter}(\{g(X_i; \Theta_j)\}_{i=1}^I)$
14: **end for**

---

# F VISUALIZATION

In Fig. 5 we provide examples visual examples of bounding boxes produced by Selective Search, our pseudo-labeled object proposal method, and SEER, specifically a ViDT+ detector trained for two stages on ImageNet. To avoid clutter, for all three methods we only include objects whose predicted bounding boxes have an IOU higher than 0.5 with an object in the ground truth set.

The images illustrate that self-training significantly improves the object discovery performance of SEER over the original region proposals. Notably, those include much smaller items, and much better performance in cluttered scenes. As stated in the main paper, this contributes to the performance of our framework and specifically the performance gains between stages.

Ground
Truth

Selective
Search

Labeled object
proposals

SEER

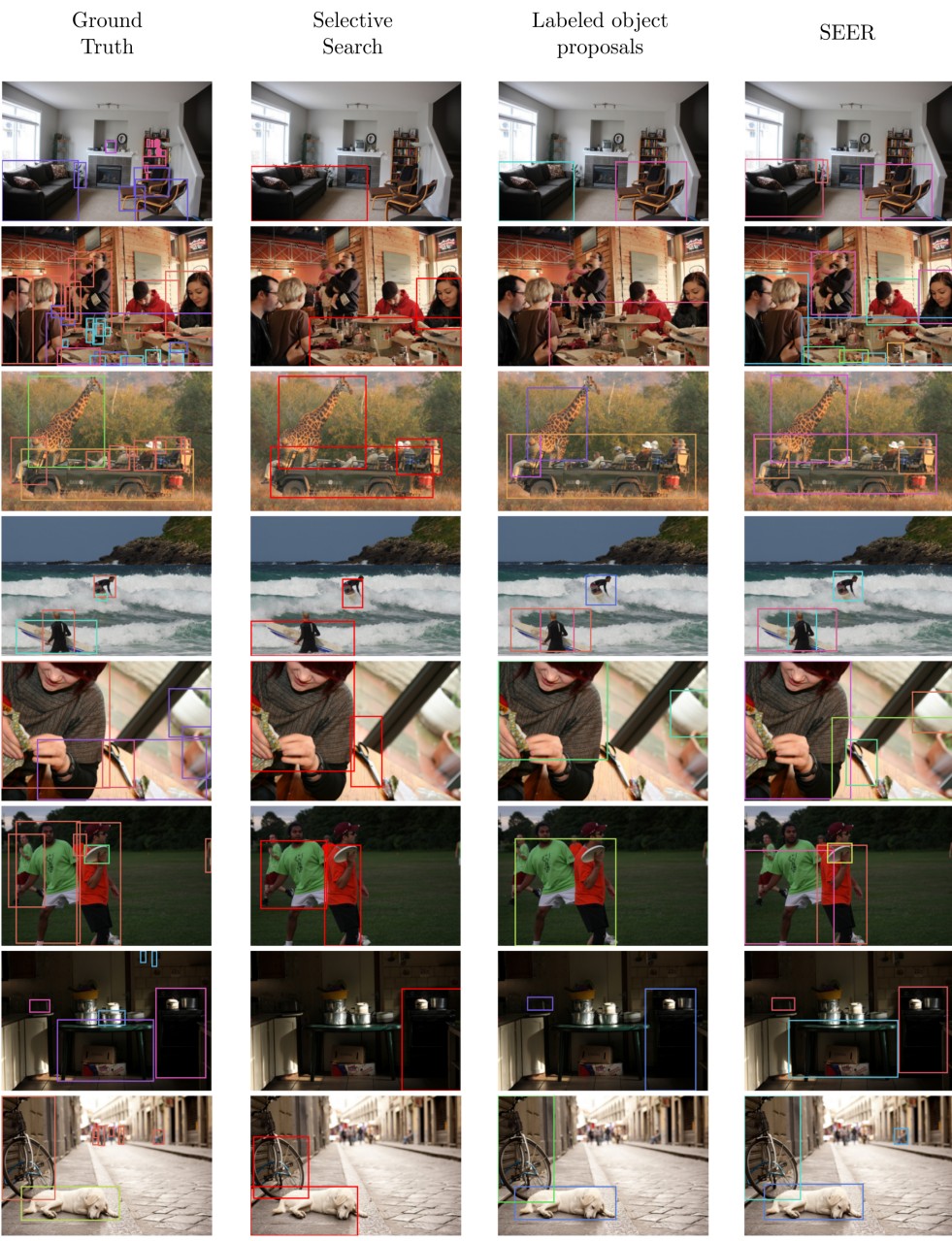

Figure 5: Examples of object proposals extracted from SEER, contrasted with the ground truth, Selective Search and our initial pseudo-labeled object proposals, extracted as described in paper Sec. 3.1. The images belong to COCO `train2017`. To avoid clutter, we only show predicted objects whose bounding boxes have an IOU greater than 0.5 with at least one ground truth object. Best seen in color.

