# OpenReview forum: "Simplifying Self-Supervised Object Detection Pretraining"
_ICLR.cc/2024/Conference — ICLR 2024 Conference Withdrawn Submission_

### Official Review · Reviewer_DE28 · 2023-10-31

**Soundness:** 3 good
**Presentation:** 3 good
**Contribution:** 2 fair
**Rating:** 5
**Confidence:** 4

**Summary:**

- This paper proposes a new method for unsupervised object detector pre-training. The main pipeline is in three steps: 1) finding object proposals in an unsupervised way; 2) clustering the proposals to form pseudo class labels ; 3) training an object detector using the proposals and their labels.

- On the downstream object detection task, the proposed method achieved comparable performance with previous methods.

**Strengths:**

1. The paper demonstrates an enhanced performance on the benchmark COCO object detection test, even if the improvement is marginal.
2. The proposed method has a good performance in data-limited scenarios, outperforming preceding methodologies substantially.
3. The manuscript is well-written, it is easy to follow, and the technical details are well explained.
4. The idea of generating a detection dataset and using it to pre-train a detector in an unsupervised way is interesting.

**Weaknesses:**

1. My main concern for this paper is its significance and potential impact for future works:
    1. While the proposed method surpasses some recent methods in terms of detection AP, the advancements are quite minimal. For instance, For example, on Mask R-CNN it only outperform CutLER by 0.3 AP. (Table 1)
    2. One anticipated advantage of unsupervised pre-training would be the ability to exponentially scale the training dataset, subsequently enhancing the model's efficacy. This paper seems to miss out on this potential. The proposed clustering of object proposals becomes increasingly complex with dataset expansion, which may not even tractable when the dataset becomes very large, i.g., web scale.
    3. Even if the clustering challenges are addressed, the method doesn't appear to capitalize on a larger dataset. For example, when switching the training set from COCO to OpenImage, which is much larger, the model performance keeps the same. (Table 6)
    4. he proposed multi-stage training only bring marginal improvement: increasing an extra stage of training brings less than 1.0 AP improvment. (Table 13)
2. The novelty of this work is also limited. The shift to generating object proposals from generating from low-level features is a simple extension of previous works. Additionally, the object detector's training mechanism seems heavily reliant on pre-existing methodologies.
3. I think the one of the intersting point of this method is its supervority on data-scarce settings (Table 4 and Table 5). However, the paper lacks in depth study about this superiority.

**Questions:**

I am wondering if the *Clustering and Classifying* paradigm can be replaced by some other methods, like contrastive learning. If this can work, it will greatly improve this work's impact.

---

> ### Author Response · Authors · 2023-11-14
> **Response to reviewer DE28**
>
> We thank reviewer RDE28 for acknowledging SEER's performance and the paper's writing, and for the issues they raised, which we address below:
>
> > While the proposed method surpasses some recent methods in terms of detection AP, the advancements are quite minimal ... & The proposed multi-stage training only bring marginal improvement ...
>
> We respectfully note that the impact of performance margins is relative to the task: across architectures SEER outperforms previous works by larger margins than they outperformed each other (Tables 1,2,3). Regarding CutLER, +0.3 AP is also the margin by which it outperforms DINO, where no detector pretraining is applied. SEER's additional +0.3 doubles the performance gains of detector pretraining.
> Similarly, we consider the gains between stages (+0.7 and +0.6 AP for VIDT+ and Def. DETR) to be significant.
> Overall, we believe that SEER's improvements are not small, and represent significant progress in the context of the task.
>
> > One anticipated advantage of unsupervised pretraining would be the ability to exponentially scale the training dataset ...
>
> Previous works on detector pretraining only pretrained on ImageNet and COCO. We have expanded on the established benchmarks by pretraining on the larger Open Images dataset, which we believe adds to our paper's contribution. We respectfully note that it is unfair to penalize our work for not conducting experiments on web-scale data, which is not done by previous works and would require immense resources. Regarding the clustering component, as mentioned in Sec. 3.1, we chose K-means specifically for its efficiency, as it can scale to billions of data points [1]. Datasets that exceed this capacity are an extreme scenario and beyond the scope of this work.
>
> > Even if the clustering challenges are addressed, the method doesn't appear to capitalize on a larger dataset...
>
> We respectfully disagree: Open Images consistently outperforms COCO (+0.3 in Table 6, +0.5 in Table 7). The fact that COCO is competitive demonstrates the sample efficiency of SEER, however these margins are significant. We will emphasize this distinction in the paper.
>
> > The novelty of this work is also limited ...
>
> SEER consists of three novel components, of which proposal extraction is one. It reaches state of the art performance across settings, and we conduct extensive experiments, that expand on previous works in terms of robustness (we test multiple detectors and pretraining datasets) and in terms of tasks we explore, producing significant insights.
> Furthermore, where previous works typically drew from object-centric representation learning methods, leading to complicated and computationally heavy frameworks, SEER outperforms them with a simple detection pipeline. Its impact, therefore, could be significant, in terms of reorienting research on detector pretraining toward developing novel techniques specific to the detection task.
> We believe the above are significant contributions and support the novelty of our work.
> Regarding proposal extraction specifically, using high level features has indeed been studied (we cover relevant works in Sec. 2). However, our approach is novel technically, distinct in motivation, and works better than previous works (SEER outperforms CutLER, the state of the art for object localization).
> Regarding the second part of the reviewer's comment, we kindly ask for clarification on the components they refer to as pre-existing and why that would be a weakness. A key feature of SEER is simplicity, as it achieves state of the art performance without the complex pipelines of previous works.
>
> > [...] one of the interesting points of this method is its superiority on data-scarce settings [...] the paper lacks in depth study about this superiority.
>
> We thank the reviewer for highlighting the performance of SEER in data-scarce settings, but we do not recognize what additional study they suggest. The experiments we conducted are as extensive as any in relevant works, and we in fact expand on them with a dedicated few-shot section in Appendix C, where we extensively analyzed SEER's behavior. We believe this comment highlights a strength of our work, not a weakness.
>
> > I am wondering if the Clustering and Classifying paradigm can be replaced by some other methods ...
>
> This component is central to SEER and we do not see how, or indeed why, it would be replaced, given SEER's state of the art results. Specifically on contrastive learning, it has already been used by works such as DETReg and Siamese DETR, which SEER outperforms while being much more efficient (contrastive frameworks require at least two forward passes per sample). However, the two approaches are not  orthogonal and future works could draw from both.
>
> We hope we have addressed the reviewer's concerns and remain eager to respond to any additional questions or comments.
>
> [1] Johnson, Jeff et al. "Billion-scale similarity search with gpus." IEEE Transactions on Big Data 7.3 (2019)

---

### Official Review · Reviewer_Yy8P · 2023-10-31

**Soundness:** 2 fair
**Presentation:** 3 good
**Contribution:** 2 fair
**Rating:** 3
**Confidence:** 3

**Summary:**

The paper addresses the challenges in object detection training, where conventional methods involve a two-phase approach: self-supervised training of the backbone followed by supervised fine-tuning using annotated data. Many existing unsupervised pretraining techniques tend to depend on low-level data, neglecting high-level class semantics, resulting in a gap between the pretraining and actual detection tasks. To tackle this issue, the authors introduce a novel framework that emphasizes semantics-based initial proposals, employs discriminative training with object pseudo-labels, and utilizes self-training. This innovative approach not only surpasses preceding techniques but also facilitates the pretraining of detectors from scratch on complex datasets, such as COCO.

**Strengths:**

1. This paper includes the experiments that pre-train on COCO, which is a good exploration. When pre-trained on COCO, the proposed method outperforms the previous methods on the linear evaluation on ImagNet.
2. The method is validated on both transformer and cnn based detectors.

**Weaknesses:**

1.	From tab 7, we can see that the models pre-trained on COCO still can not outperform the models pre-trained on ImageNet, which has been shown in the previous papers. From this point, the exploration of pre-training on COCO did not bring novelty.
2.	“Utilizing semantic information from self-supervised image encoders to produce rich object proposals and coherent pseudo-class labels” has been explored in the previous papers such as [1].
3.	In the abstract, the authors claimed that “However, existing unsupervised pretraining methods typically rely on low-level information to create pseudo-proposals that the model is then trained to localize, and ignore high-level class membership.”I do not agree with it. In fact, the Moco and Mocov2 address on the high-level semantic information, while the later works[2,3] focus low-level information and localization. So you can not say that the existing pretraining methods typically rely on low-level information. The challenge is to create a pre-training method that can balance both localization and classification.
4. I am also curious about the comparison with MAE pre-training methods.
[1] Deep Spectral Methods: A Surprisingly Strong Baseline for Unsupervised Semantic Segmentation and Localization, L. Melas-Kyriazi, C. Rupprecht, I. Laina and A. Vedaldi, Proceedings of the IEEE Conference on Computer Vision and Pattern Recognition (CVPR), 2022
[2] Fangyun Wei, Yue Gao, Zhirong Wu, Han Hu, and Stephen Lin. Aligning pretraining for detection via object-level contrastive learning. Advances in neural information processing systems.
[3] Zhenda Xie, Yutong Lin, Zhuliang Yao, Zheng Zhang, Qi Dai, Yue Cao, and Han Hu. Selfsupervised learning with swin transformers. arXiv preprint arXiv:2105.04553, 2021c.

**Questions:**

See the weakness.
I fail to discern how this work differs from prior efforts.

---

> ### Author Response · Authors · 2023-11-14
> **Response to reviewer Yy8P**
>
> We thank reviewer Yy8P for acknowledging our extensive experiments and for the issues they raised, which we address below:
>
> > [...] we can see that the models pre-trained on COCO still can not outperform the models pre-trained on ImageNet...
>
> ImageNet pretraining leads to better outcomes than COCO, as ImageNet is 10X bigger. We respectfully argue that not disproving this does not undermine the novelty of our work. Indeed, we conduct extensive experiments to study the role of dataset size and curation (Tables 6, 7). To our knowledge, ours is the first work to do this in this context, which we consider a valuable contribution rather than lack of novelty.
>
> > Utilizing semantic information [...] has been explored in the previous papers such as [1].
>
> We do not claim to be the first to propose local clustering or clustering for pseudo-labels, which have been applied in various areas in various ways, even previous to [1]. SEER is, however, the first method to use them for self-supervised detector pretraining. Regarding [1], we note that pseudo-labeling in SEER has the explicit goal of learning strong representations (hence our choice of overclustering with k=2048). Conversely, [1] sets the number of clusters to the number of classes in the dataset in order to perform unsupervised semantic segmentation, which aligns it closer to deep clustering works. Overall, we believe that our use of local and global clustering in order to pretrain powerful detectors is novel in motivation and methodology, and, combined with the other components of SEER, leads to a better and more efficient framework than previous works.
>
> > [...] the authors claimed that “However, existing unsupervised pretraining methods typically rely on low-level information to create pseudo-proposals [...] I do not agree with it ...
>
> The use of low-level vs high-level information in this context refers specifically to how most previous works on detector pretraining extract object proposals, not to contrast with backbone pretraining literature such as MoCo and [3] that use whole-of-image high-level semantic information, which is a separate subject. [2] is also a backbone pretraining method that does not train a detector (i.e. does not have a localization objective), though it includes a region proposal component. [2], as well as most previous works on detector pretraining, extract region proposals with methods that leverage low-level visual cues (e.g. Selective Search). We improve upon that, by using high-level semantic information from feature maps. Furthermore, we respectfully argue that SEER does just what the reviewer correctly identifies as a major challenge: we propose a unified framework that tackles both localization and classification, relying in both cases on high-level semantics, and in an entirely self-supervised manner.
>
> > I am also curious about the comparison with MAE ...
>
> MAE methods are used for self-supervised backbone pretraining, whereas SEER tackles self-supervised object detector pretraining. The two tasks are distinct and not directly comparable. Indeed, detector pretraining methods are typically initialized with backbones pretrained with frameworks such as MAE and MoCo.
>
> > I fail to discern how this work differs from prior efforts.
>
> As mentioned in the paper, SEER differs from previous works on detector pretraining in the following:
>
> 1. We extract object proposals using high-level semantics, rather than low-level visual cues.
> 2. We train the detector on simple class-aware detection, using pseudo-labels extracted via clustering.
> 3. We employ self-training to further improve performance.
>
> Thanks to these components, SEER: a) achieves state of the art performance across datasets and detector architectures and b) is simple and efficient, as it does use complicated objectives and does not carry the computational overhead of the student-teacher paradigm. Furthermore, our paper goes far beyond previous works experimentally. We evaluate SEER on standard benchmarks (full data, semi-supervised and few-shot), but also examine multiple datasets for pretraining, draw contrasts with unsupervised localization, and examine new benchmarks (pretraining from scratch and evaluation by linear probing), all of which yield significant insights. Finally, whereas previous works draw inspiration from self-supervised backbone pretraining methods (e.g. contrastive learning), SEER represents a new direction in that regard, as it is grounded on object detection in terms of pipeline and augmentations.
>
> We hope we have addressed the reviewer's concerns and remain eager to respond to any additional questions or comments.

---

### Official Review · Reviewer_xbku · 2023-11-01

**Soundness:** 2 fair
**Presentation:** 3 good
**Contribution:** 2 fair
**Rating:** 5
**Confidence:** 4

**Summary:**

This study proposes a new self-supervised pretraining method called SEER for object detection. SEER first generates pseudo proposals using spectral clustering of the feature map generated by a pretrained feature extractor. The generated pseudo proposals are then fed to the detector to yield an end-to-end self-supervised an object detector. The pretrained network is validated on the MS-COCO and PASCAL VOC datasets.

**Strengths:**

- The proposed framework is simple and effective. By utilizing a pretrained feature extractor to generate proposals in an unsupervised manner, this method can obtain proposals of higher quality and better semantic meaning. The proposal filtering and pseudo-class label generation also require strong engineering insights to make the pipeline work for end-to-end self-supervised object detector pretraining.

- The presentation is easy to follow, and the contribution of the paper has been made clear after comparing it with existing object detector pretraining methods in its related works section.

- The obtained results are competitive, as the method is able to achieve 46.7 AP using the Deformable DETR detector and 49.6 AP using the ViDT+ detector.

- The evaluation on different tasks, including few-shot and semi-supervised learning, and the study of different pretraining datasets are appreciated. This helps demonstrate broader significance by investigating common interesting questions.

**Weaknesses:**

- The contributions of this work may be overstated. This paper presents SEER as a unique end-to-end object detection pretraining method that can train the backbone from scratch without freezing backbone parameters. However, previous works like JoinDet and [1] have also shown the potential ability to train the backbone without freezing, so unfreezing the backbone is not an entirely new contribution.


- Moreover, the ability to train the backbone of an object detector is an overstatement to some extent. As the method still requires a pretrained backbone model to generate pseudo proposals, it then leverages the generated pseudo proposals to train its backbone feature. Given that a pretrained backbone is already provided, forcing the network to retrain a backbone from scratch is not viewed as a fully end-to-end self-pretraining method.


- Additionally, [1] has already explored the possibility of pretraining an object detector in a fully self-supervised manner without requiring an extra pretrained backbone. When comparing [1] to this study, the pipeline of [1] seems simpler and is able to train the whole model from scratch.


- The pipeline relies on clustering over a pretrained network for its pseudo proposals, which inevitably introduces many hyperparameters. For example, the number of clusters for both local and global clustering is critical to the method's performance.


References:

[1] G Jin, et al., Self-Supervised Pre-training with Transformers for Object Detection, Neurips workshop 2022. ([https://sslneurips22.github.io/paper_pdfs/paper_4.pdf](https://sslneurips22.github.io/paper_pdfs/paper_4.pdf)).

**Questions:**

- This study highlights its ability to train the backbone from scratch. I would like to raise the reverse question: How does model performance compare when using a frozen pretrained backbone versus training the backbone from scratch?

- The performance on semi-supervised results lags far behind recent studies in semi-supervised object detection [2]. What if SEER adopts the same architecture and compares results with traditional semi-supervised object detection using self-training?


References:

[2] J. Zhang, et al. Semi-DETR: Semi-Supervised Object Detection with Detection Transformers. CVPR 2023.

---

> ### Comment · Reviewer_DE28 · 2023-11-10
> **Wrong review?**
>
> Hey, I guess this review is for another paper?

---

> > ### Comment · Area_Chair_1a1r · 2023-11-11
> > **Could  you expand?**
> >
> > The review reads on topic to me.

---

> > > ### Comment · Reviewer_DE28 · 2023-11-11
> > > **Maybe caused by OpenReview problem**
> > >
> > > It's good now. But yesterday it was about another OOD paper. Maybe some bugs happened to OpenReview yesterday

---

> > > > ### Comment · Area_Chair_1a1r · 2023-11-11
> > > > **Ok thanks for flagging**
> > > >
> > > > n/a

---

> ### Author Response · Authors · 2023-11-14
> **Response to reviewer xbku**
>
> We thank reviewer xbku for their positive comments regarding our method's state of the art performance, its simplicity and effectiveness, and the writing of the paper. We further thank the reviewer for the issues they raised, which we address below:
>
> > The contributions of this work may be overstated...
>
> [1], and their follow-up work SeqCo-DETR which we cite, do not state whether they trained the backbone and mentioned they followed DETReg's initialization. So we assume that the backbone remained frozen (no code has been uploaded to allow us to verify that). We will amend the phrasing in our paper to reflect this. However, we did not claim that SEER is the first work to train the backbone, rather that most other works don't  (the exceptions being CutLER and, possibly, [1]).
>
> > Moreover, the ability to train the backbone of an object detector is an overstatement...
>
> We noted in the conclusion that SEER represents only a "promising step" toward fully self-supervised end-to-end pretraining. But please note that the pretrained backbone is used only for the extraction of pseudo-proposals. We believe that showing SEER  can effectively train from scratch (Tables 7, 8) are significant findings that we are the first to explore among the detector pretraining literature.
>
> > Additionally, [1] has already explored the possibility of pretraining an object detector...
>
> We respectfully note that [1] **does use** a pretrained backbone to initialize the detector and **does not** train the whole model from scratch, which our work is the first to attempt. Whether [1] could work without a pretrained backbone to bootstrap representations has not been shown. Furthermore, SEER consists of simple class-aware detector pretraining (with pseudo-classes), whereas [1] requires dual student-teacher architectures and a custom region matching scheme. SEER is, therefore, much more efficient and much less complex.
>
> > The pipeline relies on clustering [...], which inevitably introduces many hyperparameters...
>
> With the default hyperparameters, SEER works well (and outperforms previous works) across 3 pretraining datasets (MS COCO, ImageNet and Open Images) with 3 detector architectures (VIDT+, Def. DETR, Cascade Mask-RCNN). We therefore believe that we sufficiently demonstrate that SEER does not need extra tuning and the extra hyperparameters do not impose any burden, though tuning could further improve results on specific settings.
>
> > [...] I would like to raise the reverse question: How does model performance compare when using a frozen pretrained backbone ...
>
> We are unsure what the reviewer means. If they refer to a comparison between using a frozen backbone without detector pretraining vs. our pretraining from scratch, that is shown in Table 7. If they are referring to SEER's performance with a frozen backbone during pretraining, we examined this on VIDT+, and observed a steep performance drop at 47.9AP (vs. 49.6 when training the backbone). If the reviewer meant something else, we kindly ask them to clarify and we will eagerly respond.
>
> > The performance on semi-supervised results lags far behind recent studies in semi-supervised object detection ...
>
> For semi-supervised evaluation we closely followed the established paradigm set by previous works (DETReg, CutLER), to ensure fair comparisons. Evaluating and comparing with works that focus on semi-supervised training is outside the scope of the detector pretraining task, which focuses on how to pretrain powerful detectors, not on how best to fine-tune them in the downstream semi-supervised setting.
>
> We hope we have addressed the reviewer's concerns and remain eager to respond to any additional questions or comments.